# Attributed Graph Learning with 2-D Graph Convolution

## Abstract

Graph convolutional neural networks have demonstrated promising performance in attributed graph learning, thanks to the use of graph convolution that effectively combines graph structures and node features for learning node representations. However, one intrinsic limitation of the commonly adopted 1-D graph convolution is that it only exploits graph connectivity for feature smoothing, which may lead to inferior performance on sparse and noisy real-world attributed networks. To address this problem, we propose to explore relational information among node attributes to complement node relations for representation learning. In particular, we propose to use 2-D graph convolution to jointly model the two kinds of relations and develop a computationally efficient dimensionwise separable 2-D graph convolution (DSGC). Theoretically, we show that DSGC can reduce intra-class variance of node features on both the node dimension and the attribute dimension to facilitate learning. Empirically, we demonstrate that by incorporating attribute relations, DSGC achieves significant performance gain over state-of-the-art methods on node classification and clustering on several real-world attributed networks.

## 1 Introduction

In an attributed graph, each node is associated with a feature vector, and nodes are connected by edges encoding their relations. Commonly seen attributed graphs include citation networks where each node is a document represented by a bag-of-words feature vector and edges are citation links, web graphs where each webpage is also represented as a vector of words and edges are hyperlinks, social networks where each user is represented by a user profile vector and edges indicate user friendship, and protein-protein interaction networks where each protein is represented by a list of protein signatures and edges encode interactions between proteins. Learning on attributed graphs including node classification and clustering finds many important applications in real-world networks.

Since the connectivity patterns and node contents of an attributed graph usually contain different information, it often requires joint modelling both aspects of information to achieve good learning performance. A major class of methods (Yang et al., 2015; Pan et al., 2016; Huang et al., 2017) is devoted to learning efficient node representations of an attributed graph via nonnegative matrix factorization or random walk statistics and then perform downstream learning tasks with the learned representations. A number of semi-supervised classification methods (Belkin et al., 2006; Weston et al., 2008; Yang et al., 2016) classify nodes in an attributed graph by training a supervised classifier on node features with some kind of graph regularizer. Recently, a series of works based on graph convolutional neural networks including ChebyNet (Defferrard et al., 2016), graph convolutional networks (GCN) (Kipf & Welling, 2017), GraphSAGE (Hamilton et al., 2017) and graph attention networks (GAT) (Velickovic et al., 2018) have been shown to achieve state-of-the-art performance in node classification and clustering (Kipf & Welling, 2016; Wang et al., 2017) on attributed graphs.

The key component of these models is one-dimensional (1-D) graph convolution, a function that naturally combines graph structures and node contents by aggregating a node's features with its neighbours'. As shown in (Li et al., 2019), the graph convolutional operator used in GCN and many follow-up works acts as a low-pass graph filter that smooths a node's features with its neighbours'. Under the assumption that nearby nodes tend to be in the same class, it can produce similar feature representations for nodes in the same class, thereby making them easier to be classified or clustered. While this works well for attributed graphs with clear cluster structures, real-world networks could

be highly noisy and sparse. For example, in a web graph such as Wikipedia, a hyperlink between two webpages does not necessarily indicate that they belong to the same category, so mixing their features could be harmful for learning. Furthermore, it has been shown that many real-world networks are scale-free (Albert & Barabási, 2002), which means there exist many low-degree nodes. Since these nodes may have very few or even no links to other nodes, it would be difficult and even impossible for them to learn similar feature representations as other same-class nodes, which is an intrinsic limitation of the 1-D graph convolution that is commonly adopted in existing models.

To address these limitations, we propose to explore relational information on a different dimension – the relations between feature attributes, in addition to node relations. Likewise, a relation between two feature attributes should reflect some kind of similarity between them. The assumption is that attributes that tend to indicate same classes should have strong relations. For example, in a citation network, node attributes are words, and documents of AI category usually contain words such as "learning", "robotics", "machine", "neural", etc. These indicative words for AI category should have much stronger relations among themselves than with other non-indicative words. These informative relations can be used to construct an attribute affinity graph to smooth node features in a similar way as the node relations do, only in a different dimension. Importantly, attribute relations can complement node relations in node representation learning. For instance, consider a document that has no links to others and hence it is impossible to do feature smoothing with node relations. But with attribute relations, it can still learn similar feature representations as other same-class nodes.

In this paper, we make the following contributions.

- **Methodology**: We propose to use 2-D graph convolution to jointly model node relations and attribute relations for learning node representations of attributed graphs. Further, we develop a computationally efficient dimensionwise separable 2-D graph convolution (DS-GC), which is equivalent to performing 1-D graph convolution alternately on the node dimension and the attribute dimension respectively.

- **Theoretical Insight**: We show that regular 1-D graph convolution on the node dimension can reduce intra-class variance of node features, which explains the success of many existing methods. Further, we show that with a properly constructed attribute affinity graph, graph convolution on the attribute dimension can also reduce intra-class variance of node features. Jointly, our analysis provides a theoretical justification of DSGC.

- **Empirical Study**: We implement DSGC for node classification and clustering on attributed graphs, and compare it with state-of-the-art methods on a citation network, a web graph, and an email network. The results demonstrate the superiority of DSGC over regular 1-D graph convolution on spare and noisy real-world attributed networks. We also show that DSGC can be plugged into existing models to substantially improve their performance.

## 2 RELATED WORKS

**Structural Graph Learning.** Methods for structural graph learning only utilize graph structures (node relations). A common approach is to learn smooth low-dimensional embeddings of nodes using Markov random walks (Szummer & Jaakkola, 2002; Perozzi et al., 2014; Grover & Leskovec, 2016), Laplacian eigenmaps (Belkin & Niyogi, 2004), spectral kernels (Chapelle et al., 2003; Zhang & Ando, 2006), autoencoders (Wang et al., 2016; Cao et al., 2016; Ye et al., 2018) and generative adversarial nets (Dai et al., 2018; Wang et al., 2018). Another direction is graph-based semi-supervised classification, which includes methods based on low-density graph partition (Blum & Chawla, 2001; Blum et al., 2004; Joachims, 2003) and the popular label propagation method and its variants (Zhu et al., 2003; Zhou et al., 2004; Bengio et al., 2006; Hein & Maier, 2007; Wu et al., 2012).

**Attributed Graph Learning.** Methods for attributed graph learning take into account both graph structures and node features. A major class of methods learns node representations (Yang et al., 2015; Pan et al., 2016; Huang et al., 2017) or clustering nodes (Xia et al., 2014; Zhou et al., 2010; Li et al., 2018) by applying nonnegative matrix factorization, random walk statistics, or Laplacian eigenmaps on both graph structures and node features. Some node clustering methods use a Bayesian model (Xu et al., 2012; Bojchevski & Günnemann, 2018) or design a distance measure that trades off structural and feature information (Zhou et al., 2009; Cheng et al., 2011). Statistical relational learning methods model node relations and features with probabilistic graphical models, e.g., rela-

tional Markov networks (Taskar et al., 2002). Another category of related work is the graph-based semi-supervised node classification methods that exploit both graph structures and node features. For example, iterative classification algorithm (Sen et al., 2008) iteratively classifies an unlabeled node using its neighbours' labels and features. Manifold regularization (Belkin et al., 2006), deep semi-supervised embedding (Weston et al., 2008), and Planetoid (Yang et al., 2016) classify node features by training a supervised classifier with a Laplacian or embedding-based regularizer.

**Graph Neural Networks.** Another line of research on attributed graph learning is based on graph neural networks (Scarselli et al., 2009; Li et al., 2016). Inspired by the success of convolutional neural networks (CNN) on Euclidean data, recent works (Bruna et al., 2014; Henaff et al., 2015; Duvenaud et al., 2015; Atwood & Towsley, 2016) proposed to use 1-D graph convolution for attributed graph learning. To avoid the expensive eigen-decomposition, ChebyNet (Defferrard et al., 2016) uses a polynomial filter represented by $k$-th order polynomials of graph Laplacian via Chebyshev expansion. Graph convolutional networks (GCN) (Kipf & Welling, 2017) further simplifies ChebyNet by designing an efficient layer-wise propagation rule via a first-order approximation of spectral graph convolution. GCN achieved outstanding results in semi-supervised classification and inspired many follow-up works including MoNet (Monti et al., 2017), GraphSAGE (Hamilton et al., 2017), graph attention networks (Velickovic et al., 2018), gated attention networks (Zhang et al., 2018a), FastGCN (Chen et al., 2018b), dual graph convolutional neural network (Zhuang & Ma, 2018), stochastic GCN (Chen et al., 2018a), attributed network representation learning (Zhang et al., 2018c), LanczosNet (Liao et al., 2019), deep graph infomax (Velickovic et al., 2019), graph Markov neural networks (Qu et al., 2019), DisenGCN (Ma et al., 2019a), MixHop (Abu-El-Haija et al., 2019), etc. A recent interesting research direction is on learning graph structures for graph neural networks, e.g., Bayesian GCN (Zhang et al., 2018b) and LDS (Franceschi et al., 2019). Remarkably, LDS can automatically learn a graph for data samples even when it is not available.

Some attributed graph clustering methods based on 1-D graph convolution also showed promising performance, including graph autoencoder and graph variational autoencoder (Kipf & Welling, 2016), marginalized graph autoencoder (Wang et al., 2017), adversarially regularized graph autoencoder and adversarially regularized variational graph autoencoder (Pan et al., 2018), and adaptive graph convolution (Zhang et al., 2019). More comprehensive reviews of graph neural networks can be found in (Cai et al., 2018; Zhang et al., 2018d; Zhou et al., 2018).

## 3    2-D GRAPH CONVOLUTION

In this section, we present 2-D graph convolution for attributed graph learning. A comprehensive introduction of multi-dimensional graph Fourier transform can be found in (Kurokawa et al., 2017). Different from Kurokawa et al. (2017), here we propose a localized 2-D graph convolution to circumvent the computationally intensive graph Fourier transform. Furthermore, we propose an even simpler dimensionwise separable 2-D graph convolution to efficiently model both node relations and attribute relations along the two dimensions of the feature matrix of an attributed graph.

### 3.1    2-D GRAPH FOURIER TRANSFORM AND SPECTRAL GRAPH CONVOLUTION

A 2-D *graph signal* is a function defined on the Cartesian product of the vertex sets of two graphs. Formally, given two weighted undirected graph $\mathcal{G}^{(1)}$ and $\mathcal{G}^{(2)}$, denote the vertex sets by $\mathcal{V}^{(1)}$ and $\mathcal{V}^{(2)}$, the edge set by $\mathcal{E}^{(1)}$ and $\mathcal{E}^{(2)}$, and the weighted adjacency matrix by $\boldsymbol{A}^{(1)}$ and $\boldsymbol{A}^{(2)}$. A 2-D graph signal $x$ on $(\mathcal{G}^{(1)}, \mathcal{G}^{(2)})$ is a real-valued function $f : \mathcal{V}^{(1)} \times \mathcal{V}^{(2)} \to \mathbb{R}$, which can be conveniently represented in matrix form:

$$\boldsymbol{X} = (x_{ij}) \in \mathbb{R}^{n \times m}, \quad x_{ij} = f(\nu_i^{(1)}, \nu_j^{(2)}), \tag{1}$$

where $n = |\mathcal{V}^{(1)}|$ and $m = |\mathcal{V}^{(2)}|$. In this paper, $\mathcal{G}^{(1)}$ represents the given node graph; $\mathcal{G}^{(2)}$ represents the constructed attribute affinity graph; and $\boldsymbol{X}$ is the node feature matrix, which is a 2-D signal on the two graphs. We call the adjacency matrices $\boldsymbol{A}^{(1)}$ and $\boldsymbol{A}^{(2)}$ node affinity matrix and attribute affinity matrix respectively.

Define the graph Laplacian of $\mathcal{G}^{(1)}$ and $\mathcal{G}^{(2)}$ as $\boldsymbol{L}_l = \boldsymbol{D}^{(1)} - \boldsymbol{A}^{(1)}$ and $\boldsymbol{L}_r = \boldsymbol{D}^{(2)} - \boldsymbol{A}^{(2)}$ respectively. Denote by $\lambda_i$ and $\mu_j$ the eigenvalues of $\boldsymbol{L}_l$ and $\boldsymbol{L}_r$, and $\boldsymbol{U} = [\boldsymbol{u}_1, \cdots, \boldsymbol{u}_n]$ and $\boldsymbol{V} = [\boldsymbol{v}_1, \cdots, \boldsymbol{v}_m]$

the corresponding eigenbasis respectively, then the $n \times m$ outer products $\boldsymbol{u}_i \boldsymbol{v}_j^\top$ form a basis for the linear space $\mathbb{R}^{n \times m}$. It is known as 2-D graph Fourier basis – an analogy of the Fourier basis in classical harmonic analysis in graph domain. The corresponding eigenvalue pair $(\lambda_i, \mu_j)$ is known as the frequency of basis matrix $\boldsymbol{u}_i \boldsymbol{v}_j^\top$. Then, a 2-D graph signal $\boldsymbol{X}$ can be decomposed as:

$$\boldsymbol{X} = \sum_{ij} s_{ij}(\boldsymbol{u}_i \boldsymbol{v}_j^\top) = \boldsymbol{U} \boldsymbol{S} \boldsymbol{V}^\top, \; \boldsymbol{S} = (s_{ij}) \in \mathbb{R}^{n \times m}. \tag{2}$$

Then, we can define 2-D *graph Fourier transform* as $\boldsymbol{S} = \boldsymbol{U}^\top \boldsymbol{X} \boldsymbol{V}$, where $\boldsymbol{S}$ is called the spectrum of signal $\boldsymbol{X}$ or Fourier coefficients.

Based on 2-D graph Fourier transform, we can now manipulate 2-D graph signals in the spectral (frequency) domain and define 2-D spectral graph convolution. By the convolution theorem, the convolution of two signals equals to point-wise multiplication of their spectrum in the spectral domain. 2-D *spectral graph convolution* is a function conv : $\mathcal{R}^{n \times m} \to \mathcal{R}^{n \times m}$ that takes signal $\boldsymbol{X}$ as input and outputs a new signal $\boldsymbol{Z}$:

$$\boldsymbol{Z} = \sum_{ij} p(\lambda_i, \mu_j) s_{ij}(\boldsymbol{u}_i \boldsymbol{v}_j^\top) = \boldsymbol{U}(\boldsymbol{S} \circ \boldsymbol{P})\boldsymbol{V}^\top, \tag{3}$$

where $p(\lambda, \mu) : \mathbb{R} \times \mathbb{R} \to \mathbb{R}$ is the frequency response of the convolution; $\circ$ is Hadamard (element-wise) product; and $\boldsymbol{P} \in \mathbb{R}^{n \times m}$ with $P_{ij} = p(\lambda_i, \mu_j)$ is the frequency response in matrix form.

## 3.2 Fast Localized 2-D Spatial Graph Convolution

Although Eq. (3) well defines 2-D graph convolution, it is often impractical to perform convolution in the spectral domain, due to the high cost of computing the eigenbasis $\boldsymbol{U}, \boldsymbol{V}$ needed for Fourier transform. Similar to (Defferrard et al., 2016) on 1-D graph convolution, here we propose 2-D spatial graph convolution to avoid intensive computation. Without loss of generality, we restrict the frequency response $p(\cdot, \cdot)$ to be a polynomial of two variables $\lambda, \mu$ with parameters $\boldsymbol{\Theta} \in \mathbb{R}^{n \times m}$, i.e., $p(\lambda, \mu) = \sum_{i,j} \theta_{ij} \lambda^i \mu^j$. Then, the 2-D spectral graph convolution in Eq. (3) becomes

$$\boldsymbol{Z} = \sum_{i=0}^{n-1} \sum_{j=0}^{m-1} \theta_{ij} \boldsymbol{L}_l^i \boldsymbol{X} \boldsymbol{L}_r^j. \tag{4}$$

Eq. (4) is called 2-D *spatial graph convolution*, as it manipulates the signal $\boldsymbol{X}$ in the spatial domain. Parameter $\boldsymbol{\Theta}$ is called the kernel of the convolution. Here, the spatial convolutional filter is localized. Denote by $k_1$ and $k_2$ the largest exponent of $\lambda$ and $\mu$ in the polynomial $p$ respectively, then $i > k_1$ and $j > k_2$ imply $\theta_{ij} = 0$. The convoluted signal $z_{ij}$ of vertex pair $(\nu_i^{(1)}, \nu_j^{(2)})$ only depends on the neighbourhood of $\nu_i^{(1)}$ within $k_1$ hops and the neighbourhood of $\nu_j^{(2)}$ within $k_2$ hops, so the filter is said to be $k_1$-localized on $\mathcal{G}^{(1)}$ and $k_2$-localized on $\mathcal{G}^{(2)}$, and the size of the kernel $\boldsymbol{\Theta}$ is $k_1 \times k_2$.

**Dimensionwise Separable 2-D Graph Convolution (DSGC)** Although the above spatial graph convolution avoids the computationally expensive Fourier transform, its general form with kernel size $k_1 \times k_2$ still involves at least $k_1 \times k_2$ matrix multiplications. Inspired by the depthwise separable convolution proposed in (Howard et al., 2017), we streamline spatial graph convolution by restricting the rank of $\boldsymbol{\Theta}$ to be one. Consequently, $\boldsymbol{\Theta}$ is decomposed as an outer product of two vectors $\boldsymbol{\theta}^{(1)} \in \mathbb{R}^n$ and $\boldsymbol{\theta}^{(2)} \in \mathbb{R}^m$. The frequency response $p$ can be decomposed as a product of two single variable polynomials, i.e., $p(\lambda, \mu) = p_1(\lambda)p_2(\mu)$, where $p_1(\lambda) = \sum_i \theta_i^{(1)} \lambda^i$ and $p_2(\mu) = \sum_j \theta_j^{(2)} \mu^j$. Finally, the 2-D spatial graph convolution in Eq. (4) becomes

$$\boldsymbol{Z} = \boldsymbol{G} \boldsymbol{X} \boldsymbol{F}, \quad \text{where } \boldsymbol{G} = p_1(\boldsymbol{L}_l) \text{ and } \boldsymbol{F} = p_2(\boldsymbol{L}_r). \tag{5}$$

We call Eq. (5) dimensionwise separable graph convolution (DSGC). The fastest way to compute it only requires $k_1 + k_2$ matrix multiplications, much less than the $k_1 \times k_2$ matrix multiplications needed by a general 2-D spatial graph convolution.

We call $\boldsymbol{G} \boldsymbol{X}$ node graph convolution and $\boldsymbol{G}$ the node graph convolutional filter. Similarly, we call $\boldsymbol{X} \boldsymbol{F}$ attribute graph convolution and $\boldsymbol{F}$ the attribute graph convolutional filter. Notably, $\boldsymbol{G}$ and $\boldsymbol{F}$ can complement each other to learn better node representations. As illustrated in Fig. 1, the node affinity

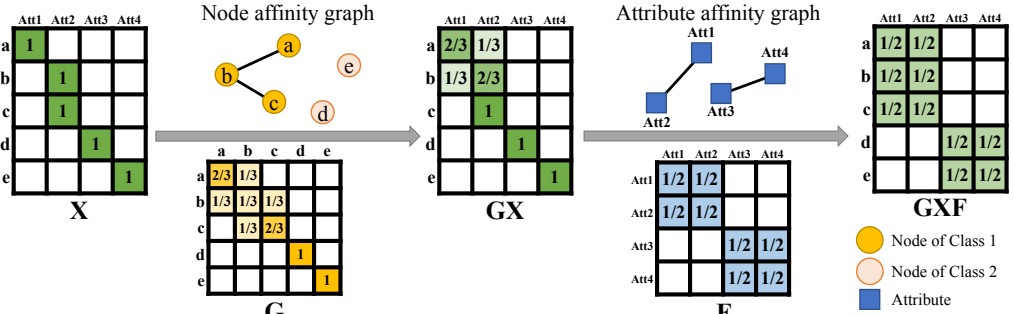

Figure 1: Illustration of DSGC.

graph only captures node relations of Class 1, and applying node graph convolution on the feature matrix $X$ smooths the node features of Class 1, but it does not affect those of Class 2. Fortunately, the attribute affinity graph encodes complementary relational information on another dimension, and further applying attribute graph convolution makes the node features of Class 2 similar and those of Class 1 even more similar, thereby making them much easier to be classified or clustered.

## 4 INTRA-CLASS VARIANCE REDUCTION BY DSGC

Given a data distribution, the lowest possible error rate an classifier can achieve is the Bayes error rate (Fukunaga, 2013), which is caused by the intrinsic overlap between different classes and cannot be avoided. In this section, we show that DSGC with proper graph convolutional filters can reduce intra-class variance of the data distribution while keeping class centers roughly unchanged, hence reducing the overlap between classes and improving learning performance.

**Intra-class Variance and Inter-class Variance.** Suppose samples $x_i$ and their labels $y_i$ are observations of a random vector $\mathbb{X} = [\mathbb{X}_1, \cdots, \mathbb{X}_m]^\top$ and a random variable $\mathbb{Y}$ respectively. We define the variance of random vector $\mathbb{X}$ to be the sum of the variance of each dimension $\mathbb{X}_j$, i.e., the trace of the covariance matrix of $\mathbb{X}$. According to law of total variance (Grinstead & Snell, 2012), the variance of $\mathbb{X}$ can be divided into intra-class variance and inter-class variance:

$$\mathrm{Var}\left(\mathbb{X}\right) = \underbrace{\mathrm{E}\left[\mathrm{Var}\left(\mathbb{X}|\mathbb{Y}\right)\right]}_{\text{Intra-class Variance}} + \underbrace{\mathrm{Var}\left(\mathrm{E}\left[\mathbb{X}|\mathbb{Y}\right]\right)}_{\text{Inter-class Variance}}, \tag{6}$$

where the conditional variance $\mathrm{Var}\left(\mathbb{X}|\mathbb{Y}=k\right)$ is the variance of class $k$ and the conditional expectation $\mathrm{E}\left[\mathbb{X}|\mathbb{Y}=k\right]$ is the $k$-th class center. Intra-class variance (IntraVar) measures the average divergence within each class, while inter-class variance (InterVar) measures the divergence among class centers. We are interested in the IntraVar/InterVar ratio. Here, we assume that each class of data $\mathrm{Pr}\left(\mathbb{X}|\mathbb{Y}=k\right)$ follows a unimodal distribution (e.g. Gaussian, chi-square, Laplace), a class of most common distributions in the real world, and with roughly convex contours. Under this assumption, a low IntraVar/InterVar ratio generally indicates low classification error, since the class overlap is reduced. This is also verified by our experiments on real attributed networks in section 7.

### 4.1 INTRA-CLASS VARIANCE REDUCTION BY NODE GRAPH CONVOLUTION

As variance of sample mean is always less than variance of individual samples, averaging samples of the same class can always reduce intra-class variance. Actually this is how node graph convolution works. For any node $i$, node graph convolution $GX$ produces a new feature vector $z_i = \sum_j G_{ij} x_j$. When $G$ is a stochastic matrix, the output feature vector $z_i$ is a weighted average of the neighbours of $x_i$. Denote by $\mathbb{Z}$ a random vector of $z_i$. Intuitively, as long as each node $i$ has enough same-class neighbours, $\mathbb{Z}$ will have a smaller IntraVar/InterVar ratio than $\mathbb{X}$. Formally, assume that the graph is generated by a stochastic block model, where nodes from the same class are connected with probability $p$ and those from different classes are connected with probability $q$. Then, with the stochastic graph filter $G = D^{-1}A^{(1)}$, we have the following theorem.

**Theorem 1.** *When $q$ is sufficiently small, the* IntraVar/InterVar *ratio of $\mathbb{Z}$ is less than or equal to that of $\mathbb{X}$, i.e.,*

$$\frac{\mathrm{E}\left[\mathrm{Var}\left(\mathbb{Z}|\mathbb{Y}\right)\right]}{\mathrm{Var}\left(\mathrm{E}\left[\mathbb{Z}|\mathbb{Y}\right]\right)} \leq \frac{\mathrm{E}\left[\mathrm{Var}\left(\mathbb{X}|\mathbb{Y}\right)\right]}{\mathrm{Var}\left(\mathrm{E}\left[\mathbb{X}|\mathbb{Y}\right]\right)}. \tag{7}$$

The proof is given in the Appendices. Under the assumption that nodes in the same class are most likely to be connected, node graph convolution $\boldsymbol{G}$ can reduce intra-class variance while keeping inter-class variance roughly unchanged, thereby decreasing the IntraVar/InterVar ratio.

### 4.2 INTRA-CLASS VARIANCE REDUCTION BY ATTRIBUTE GRAPH CONVOLUTION

In the following, we show that a proper attribute graph convolutional filter $\boldsymbol{F}$ can also reduce the IntraVar/InterVar ratio. We use the convention that the random vector $\mathbb{X}$ is a column vector, and hence the attribute graph convolution $\boldsymbol{X}\boldsymbol{F}$ results in a new random vector $\boldsymbol{F}^\top\mathbb{X}$. We also assume that the node features are mean-centered, i.e. $\mathrm{E}\left[\mathbb{X}\right] = \boldsymbol{0}$. Denote by $\mathrm{Cov}^{(k)} = \mathrm{Cov}\left(\mathbb{X}|\mathbb{Y} = k\right)$ the covariance matrix of $\mathbb{X}$ w.r.t. each class, and denote by $\pi_k = \mathrm{Pr}(\mathbb{Y} = k)$ the portion of each class. Then, after attribute graph convolution, the intra-class variance of data becomes

$$\mathrm{E}\left[\mathrm{Var}\left(\boldsymbol{F}^\top\mathbb{X}|\mathbb{Y}\right)\right] = \sum_k \pi_k \sum_{ij} \mathrm{Cov}_{ij}^{(k)}(\boldsymbol{F}\boldsymbol{F}^\top)_{ij} \tag{8}$$

**Theorem 2.** *If the attribute graph convolutional filter $\boldsymbol{F}$ is a doubly stochastic matrix, then the output of attribute graph convolution has an intra-class variance less than or equal to that of $\mathbb{X}$, i.e.,*

$$\sum_i F_{ij} = \sum_j F_{ij} = 1 \text{ and } F_{ij} \geq 0, \forall\, i, j \quad \Rightarrow \quad \mathrm{E}\left[\mathrm{Var}\left(\boldsymbol{F}^\top\mathbb{X}|\mathbb{Y}\right)\right] \leq \mathrm{E}\left[\mathrm{Var}\left(\mathbb{X}|\mathbb{Y}\right)\right].$$

The proofs of Eq. (8) and Theorem 2 are given in the Appendices. Eq. (8) suggests that to reduce intra-class variance, the attribute affinity graph should connect attributes $\mathbb{X}_i$ and $\mathbb{X}_j$ with small or even negative covariance $\mathrm{Cov}_{ij}^{(k)}$. For example, in Figure 1, the attributes Att3 and Att4 are both indicative of class 2, but negatively correlated w.r.t. class 2, so connecting them in the attribute affinity graph can greatly reduce the intra-class variance of class 2.

To achieve a low IntraVar/InterVar ratio, in addition to reducing intra-class variance, we also need to keep the class centers apart after convolution, which then depends on the quality of the attribute affinity graph. A good attribute affinity graph should connect attributes that share similar expectations conditioned on $\mathbb{Y}$. Formally, each attribute $\mathbb{X}_j$ has $K$ conditional expectations w.r.t. $\mathbb{Y}$, which are denoted as a vector $\boldsymbol{e}_j = (\mathrm{E}\left[\mathbb{X}_j|\mathbb{Y} = 1\right], \cdots, \mathrm{E}\left[\mathbb{X}_j|\mathbb{Y} = K\right]) \in \mathbb{R}^K$. We have the following.

**Theorem 3.** *If $\forall F_{ij} \neq 0$, $\|\boldsymbol{e}_i - \boldsymbol{e}_j\|_2 \leq \varepsilon$, then the distance between $\boldsymbol{e}_j$ and $\widehat{\boldsymbol{e}}_j = \sum_i F_{ij}\boldsymbol{e}_i$ is also less than or equal to $\varepsilon$, i.e.,*

$$\|\boldsymbol{e}_i - \boldsymbol{e}_j\|_2 \leq \varepsilon,\ \forall F_{ij} \neq 0 \quad \Rightarrow \quad \|\boldsymbol{e}_j - \widehat{\boldsymbol{e}}_j\|_2 \leq \varepsilon,$$

*and $\varepsilon$ can be arbitrarily small with a proper $\boldsymbol{F}$.*

By Theorem 3, since the conditional expectations of each attribute may change little after attribute graph convolution, one can infer that the class centers will also change little, and so does the inter-class variance. Combining Theorems 2 & 3, it suggests that a proper attribute affinity graph should connect attributes that have similar class means but are less positively correlated, so as to achieve a low IntraVar/InterVar ratio and improve performance. Again, it can be seen in Figure 1 that the attributes Att3 and Att4 have exactly the same class means, but are negatively correlated w.r.t. class 2, and connecting them in the attribute affinity graph reduces the intra-class variance of class 2 to 0.

## 5 ATTRIBUTED GRAPH LEARNING WITH DSGC

### 5.1 UNSUPERVISED NODE REPRESENTATION LEARNING

Given an attributed graph with node feature matrix $\boldsymbol{X}$, we can learn the node representations $\boldsymbol{Z}$ in an unsupervised manner by applying DSGC on $\boldsymbol{X}$, i.e.,

$$\boldsymbol{Z} = \boldsymbol{G}\boldsymbol{X}\boldsymbol{F}, \tag{9}$$

and then perform various downstream learning tasks with $\boldsymbol{Z}$.

**Node Classification.** With the learned node representations $\boldsymbol{Z}$, we can simply train a classifier such as multilayer perceptron with the labeled nodes, and then apply the trained classifier on the unlabeled nodes to predict their labels.

**Node Clustering.** With the learned node representations $\boldsymbol{Z}$, we propose to cluster nodes as follows. We first apply a linear kernel on $\boldsymbol{Z}$ to learn pairwise proximity between nodes, i.e., $\boldsymbol{K} = \boldsymbol{Z}\boldsymbol{Z}^{\top}$, and then perform spectral clustering (Perona & Freeman, 1998; Von Luxburg, 2007) on $\boldsymbol{K}$.

### 5.2 END-TO-END SEMI-SUPERVISED LEARNING

For semi-supervised learning on attributed graphs, we can also use DSGC to replace the 1-D graph convolution used in existing end-to-end semi-supervised learning models including GCN (Kipf & Welling, 2017), GAT (Velickovic et al., 2018) and GraphSAGE (Hamilton et al., 2017). For example, to incorporate DSGC into the vanilla GCN, we can modify the first layer propagation of GCN as:

$$\boldsymbol{H}^{(1)} = \sigma(\boldsymbol{G}\boldsymbol{X}\boldsymbol{F}\boldsymbol{W}^{(1)}), \tag{10}$$

where $\boldsymbol{H}^{(1)}$ is the hidden units in the first layer, $\boldsymbol{W}^{(1)} \in \mathbb{R}^{m \times l}$ is the trainable parameters of GCN, and $\sigma$ is a nonlinear function such as ReLU. Both $\boldsymbol{G}$ and $\boldsymbol{F}$ are fixed filters, where $\boldsymbol{G}$ is the node graph convolutional filter of GCN, and $\boldsymbol{F}$ is the proposed attribute graph convolutional filter.

Importantly, Eq. (10) can be considered as feeding a filtered feature matrix $\boldsymbol{X}\boldsymbol{F}$ instead of the raw feature matrix $\boldsymbol{X}$ to GCN. By our above analysis, a proper attribute graph filter $\boldsymbol{F}$ can reduce intra-class variance, which makes $\boldsymbol{X}\boldsymbol{F}$ much easier to classify and guarantees to help train a better model. Further, it can be shown that the model trained by Eq. (10) is essentially different from that of GCN almost surely. For GCN, the model is freely chosen from the parameter space $\boldsymbol{W}^{(1)}$, while the model trained by Eq. (10) is restricted in a subspace $\boldsymbol{F}\boldsymbol{W}^{(1)}$. Since $\boldsymbol{F}$ is low-pass (section 6.2), it is also low-rank, and $\boldsymbol{F}\boldsymbol{W}^{(1)}$ is a small subspace of $\mathbb{R}^{m \times l}$ projected by $\boldsymbol{F}$. Model parameters in this subspace are generally better in terms of the generalization performance (test accuracy), due to the variance reduction property of $\boldsymbol{F}$. However, the model learned by Eq. (10) can hardly be learned by GCN, since the subspace $\boldsymbol{F}\boldsymbol{W}^{(1)}$ has measure zero, which is an extremely small subset of $\mathbb{R}^{m \times l}$.

## 6 IMPLEMENTATION OF DSGC

### 6.1 NODE GRAPH CONVOLUTIONAL FILTERS

The node affinity graph ($\boldsymbol{A}^{(1)}$) is given as part of the dataset. There are various graph convolutional filters available (Li et al., 2019), e.g., the one used in GCN (Kipf & Welling, 2017) is a symmetrically normalized node affinity matrix. In our experiments, we use a row normalized node affinity matrix (consistent to our analysis in section 4.1) of order 2 (following GCN) as the filter for node graph convolution:

$$\boldsymbol{G} = (\boldsymbol{D}_1^{-1}\boldsymbol{A}^{(1)})^2, \tag{11}$$

where $\boldsymbol{D}_1$ is the degree matrix of $\boldsymbol{A}^{(1)}$. We observe in our experiments that the performance of the symmetrically normalized node affinity matrix (GCN filter) is similar.

### 6.2 ATTRIBUTE GRAPH CONVOLUTIONAL FILTERS

A key issue in implementing DSGC is to construct the attribute affinity graph ($\boldsymbol{A}^{(2)}$). Possible ways of constructing $\boldsymbol{A}^{(2)}$ include extracting entity relation information from existing knowledge bases, building a similarity graph from features, or identifying correlations by domain knowledge. In the following, we describe two methods for text data, which have been proven useful by our experiments.

**Positive point-wise mutual information (PPMI).** Positive PMI (Church & Hanks, 1990) is a common tool for measuring the association between two words in natural language processing. Given a pair of words $w_i$ and $w_j$, the edge weight is defined as the PPMI between $w_i$ and $w_j$:

$$a_{ij}^{(2)} = \text{PPMI}(w_i, w_j) = \left[\log \frac{p(w_i, w_j)}{p(w_i)p(w_j)}\right]_+, \tag{12}$$

where $p(w_i, w_j)$ and $p(w_i)$ are learned by sliding a window over a large corpus of text. PMI reflects word collocation, as it assumes that if two words co-occur more than expected under independence, there must be some kind of semantic relation between them, which is often true in practice. With the constructed attribute affinity graph $\boldsymbol{A}^{(2)}$ by PPMI, in our experiments, we use a symmetrically normalized affinity matrix as the filter:

$$\boldsymbol{F} = \boldsymbol{D}_2^{-\frac{1}{2}} \boldsymbol{A}^{(2)} \boldsymbol{D}_2^{-\frac{1}{2}}, \tag{13}$$

where $\boldsymbol{D}_2$ is the degree matrix of $\boldsymbol{A}^{(2)}$.

**Word embedding based $k$-NN graphs.** Word embedding is a collection of techniques that map vocabularies to vectors in a Euclidean space. Embeddings of words are pre-trained vectors learned from corpus with algorithms such as GloVe (Pennington et al., 2014). Since word embeddings capture semantic relations between words (Bakarov, 2018), they can be used for constructing an attribute affinity graph. With the embedding vectors, we can construct a $k$-NN graph with some proximity metric such as the Euclidean distance. With the constructed attribute affinity graph $\boldsymbol{A}^{(2)}$, in our experiments, we use the following one-step lazy random walk filter (Li et al., 2019):

$$\boldsymbol{F} = (\boldsymbol{I} + \boldsymbol{D}_2^{-\frac{1}{2}} \boldsymbol{A}^{(2)} \boldsymbol{D}_2^{-\frac{1}{2}})/2, \tag{14}$$

where $\boldsymbol{D}_2$ is the degree matrix of $\boldsymbol{A}^{(2)}$. Note that the filters in Eq. (13) and (14) are slightly different, since the diagonal weight of the affinity matrix constructed by word embedding is much smaller than that of the PPMI affinity matrix, and we want to assign sufficient weight to each attribute itself. Also, we use filters of order 1 since the PPMI affinity matrix is very dense. According to the analysis in Li et al. (2019), Eq. (13) and (14) are both low-pass filters.

## 7 EMPIRICAL STUDY

To validate the effectiveness of DSGC, we conduct extensive experiments for semi-supervised node classification and node clustering on three large real-world attributed networks including 20 Newsgroups (20 NG) (Lang, 1995), Wikipeedia (Wiki) (West et al., 2009; West & Leskovec, 2012) and Large Cora (L-Cora) (McCallumzy et al., 1999; Li et al., 2019). [1] Due to space limitation, details of datasets, experimental setup and computational time are provided in the Appendices.

**Variance Reduction.** First of all, to verify our analysis in section 4, we demonstrate the variance reduction effect of both node graph convolution and attribute graph convolution. As shown in Figure 2, 1-D graph convolution with either $\boldsymbol{G}$ or $\boldsymbol{F}$ already greatly reduces the IntraVar/InterVar ratio, and together they further significantly reduce the ratio.

### 7.1 NODE CLASSIFICATION

**Baselines.** We test the proposed node classification method by DSGC with or without using the node graph convolutional filter ($\boldsymbol{G}$) and the attribute graph convolutional filter ($\boldsymbol{F}$) in five cases. We use a two-layer multilayer perceptron as the classifier for DSGC. PPMI and Emb denote constructing the attribute affinity graph with positive point-wise mutual information and word embedding respectively. We compare DSGC with the following baselines: label propagation (LP) (Wu et al., 2012), multi-layer perceptron (MLP), GCN (Kipf & Welling, 2017), generalized label propagation (GLP) (Li et al., 2019), GraphSAGE (Hamilton et al., 2017), graph attention networks (GAT) (Velickovic et al., 2018), and deep graph infomax (DGI) (Velickovic et al., 2019). We also try to incorporate DSGC into GCN, GAT, GraphSAGE, and LDS (Franceschi et al., 2019) (see the experimental setup of LDS in Appendix A.4) as described in section 5.2 to improve their performance.

**Performance.** We test semi-supervised node classification under two scenarios – 20 labels per class and 5 labels per class. For each task, we report the mean classification accuracies and standard deviations over 10 runs, as summarized in Table 1, where the top 2 accuracies are highlighted in bold. In Table 2, we report the results of GAT, GCN, GraphSAGE, and LDS after incorporating DSGC. The following observations can be made.

---

[1]Note that we did not use the "Cora", "Citeseer" and "PubMed" datasets as in (Kipf & Welling, 2017; Yang et al., 2016; Sen et al., 2008), since the attribute (word) lists are not provided in these datasets.

Table 1: Classification accuracy.

| Datasets | | | 20 NG | | L-Cora | | Wiki | |
|---|---|---|---|---|---|---|---|---|
| Methods | $G$ | $F$ | 20 labels | 5 labels | 20 labels | 5 labels | 20 labels | 5 labels |
| LP | ✓ | ✗ | $16.39 \pm 0.20$ | $8.62 \pm 0.20$ | $55.77 \pm 0.97$ | $38.97 \pm 3.15$ | $9.53 \pm 0.05$ | $10.54 \pm 0.19$ |
| MLP | ✗ | ✗ | $65.77 \pm 0.22$ | $36.10 \pm 1.11$ | $51.05 \pm 0.71$ | $33.56 \pm 2.43$ | $60.86 \pm 0.69$ | $29.95 \pm 1.04$ |
| GLP | ✓ | ✗ | $76.21 \pm 0.18$ | $47.86 \pm 1.63$ | $67.58 \pm 1.06$ | $51.04 \pm 1.61$ | $33.42 \pm 1.44$ | $15.38 \pm 1.37$ |
| GCN | ✓ | ✗ | $76.14 \pm 0.24$ | $47.70 \pm 1.64$ | $66.69 \pm 0.98$ | $48.62 \pm 1.81$ | $48.65 \pm 0.65$ | $38.30 \pm 1.48$ |
| GAT | ✓ | ✗ | $75.16 \pm 0.25$ | $49.14 \pm 1.55$ | $66.49 \pm 1.01$ | $49.27 \pm 2.25$ | $48.88 \pm 0.68$ | $36.90 \pm 1.75$ |
| DGI | ✓ | ✗ | $73.34 \pm 0.27$ | $\mathbf{66.57} \pm 0.63$ | $61.39 \pm 0.50$ | $\mathbf{54.77} \pm 1.24$ | $49.70 \pm 1.63$ | $43.64 \pm 1.89$ |
| GraphSAGE | ✓ | ✗ | $65.73 \pm 0.17$ | $42.48 \pm 0.77$ | $57.28 \pm 0.71$ | $46.79 \pm 1.91$ | $65.52 \pm 0.62$ | $48.81 \pm 0.76$ |
| DSGC ($GX$) | ✓ | ✗ | $76.27 \pm 0.20$ | $47.92 \pm 1.57$ | $67.16 \pm 1.04$ | $51.93 \pm 1.46$ | $49.78 \pm 0.50$ | $43.79 \pm 1.48$ |
| DSGC ($XF$) | ✗ | Emb | $68.08 \pm 0.22$ | $43.41 \pm 0.99$ | $55.34 \pm 0.66$ | $38.13 \pm 2.20$ | $\mathbf{68.43} \pm 0.37$ | $\mathbf{54.25} \pm 0.99$ |
| | ✗ | PPMI | $76.10 \pm 0.21$ | $61.45 \pm 0.74$ | $58.54 \pm 0.79$ | $44.58 \pm 2.00$ | $\mathbf{69.53} \pm 0.36$ | $\mathbf{58.44} \pm 1.48$ |
| DSGC ($GXF$) | ✓ | Emb | $\mathbf{77.38} \pm 0.14$ | $55.24 \pm 1.21$ | $\mathbf{68.38} \pm 0.85$ | $52.87 \pm 1.56$ | $58.66 \pm 0.65$ | $45.71 \pm 1.64$ |
| | ✓ | PPMI | $\mathbf{81.91} \pm 0.20$ | $\mathbf{70.35} \pm 0.72$ | $67.60 \pm 0.82$ | $54.07 \pm 1.13$ | $58.35 \pm 0.52$ | $46.72 \pm 1.66$ |

$^\star$ ✓ and ✗ indicate using/not using $G$ or $F$.

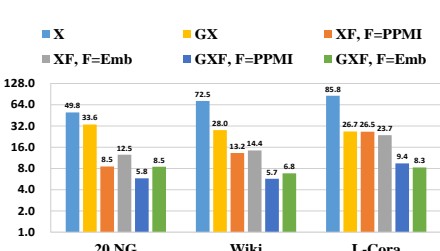

Figure 2: IntraVar/InterVar ratios.

Table 2: Improve baselines with DSGC.

| Methods | $F$ | 20 NG | L-Cora | Wiki |
|---|---|---|---|---|
| GAT | ✗ | $75.16 \pm 0.25$ | $66.49 \pm 1.01$ | $48.88 \pm 0.68$ |
| | PPMI | $80.52 \pm 0.16$ | $67.00 \pm 0.65$ | $55.51 \pm 0.38$ |
| GCN | ✗ | $76.14 \pm 0.24$ | $66.69 \pm 0.98$ | $48.65 \pm 0.65$ |
| | PPMI | $81.80 \pm 0.22$ | $65.83 \pm 0.87$ | $60.22 \pm 0.71$ |
| LDS | ✗ | $75.74 \pm 0.68$ | $62.51 \pm 0.70$ | $62.89 \pm 0.61$ |
| | PPMI | $81.98 \pm 0.46$ | $62.94 \pm 0.32$ | $66.97 \pm 0.63$ |
| GraphSAGE | ✗ | $65.73 \pm 0.17$ | $57.28 \pm 0.71$ | $65.52 \pm 0.62$ |
| | PPMI | $76.27 \pm 0.33$ | $60.23 \pm 1.81$ | $67.26 \pm 0.52$ |

$^\star$ ✗ indicates not using $F$.

**1)** Node graph convolution does not always help. On 20 NG and L-Cora, methods based on node graph convolution such as DSGC ($GX$), GLP, GCN and GAT all outperform MLP significantly. However, on Wiki, node graph convolution harms the performance, which is due to the highly noisy hyperlink graph with intra-class edge ratio 38.0%, much lower than that of 20 NG (96.8%) and L-Cora (76.5%) (see Table 5 in the Appendices). This shows the limitation of node graph convolution.

**2)** Attribute graph convolution works. On all the three datasets, DSGC ($XF$) outperforms MLP by a large margin. This shows that attribute graph convolution can learn useful node representations even when there are no connections between nodes. Remarkably, on Wiki, where the hyperlink graph is of bad quality, DSGC ($XF$) with either PPMI or Emb outperforms all the baselines by a large margin. DSGC ($XF$) with PPMI improves upon the best baseline by 4% and 10% in absolute accuracy for tasks with 20 labels and 5 labels per class respectively.

**3)** 2-D graph convolution is effective.

• For datasets with good node graphs such as 20 NG and L-Cora, DSGC ($GXF$) performs much better than either DSGC ($GX$) or DSGC ($XF$). DSGC ($GXF$) with PPMI achieves the best performance. On 20 NG, it improves upon the best baseline by more than 5% for tasks with 20 labels per class, and about 4% for tasks with 5 labels per class, both in absolute accuracy. On L-Cora, it is comparable with the best baseline. On both datasets, for tasks with 20 labels per class, DSGC ($GXF$) with Emb also outperforms all the baselines; for tasks with 5 labels per class, it outperforms all the baselines except DGI. This shows that node graph convolution and attribute graph convolution can complement each other and lead to significant performance gain.

• For datasets with bad node graphs such as Wiki, DSGC ($GXF$) significantly improves upon DSGC ($GX$) and outperforms all the baselines except GraphSAGE, which uses extra labeled data for validation while DSGC uses none. Nevertheless, since node graph convolution degrades performance on such a noisy graph, DSGC ($XF$) performs better than DSGC ($GXF$).

Table 3: Clustering performance.

| Datasets | | | 20 NG | | L-Cora | | Wiki | |
|---|---|---|---|---|---|---|---|---|
| Methods | $G$ | $F$ | Acc(%) | NMI(%) | Acc(%) | NMI(%) | Acc(%) | NMI(%) |
| Spectral | ✗ | ✗ | $25.29_{\pm 1.01}$ | $28.18_{\pm 0.74}$ | $28.22_{\pm 1.01}$ | $11.61_{\pm 0.04}$ | $29.25_{\pm 0.00}$ | $21.83_{\pm 0.00}$ |
| GAE | ✓ | ✗ | $38.92_{\pm 1.39}$ | $44.58_{\pm 0.40}$ | $34.45_{\pm 0.76}$ | $22.38_{\pm 0.18}$ | $33.78_{\pm 0.32}$ | $22.88_{\pm 0.20}$ |
| VGAE | ✓ | ✗ | $25.04_{\pm 0.81}$ | $25.72_{\pm 0.77}$ | $29.45_{\pm 1.25}$ | $17.53_{\pm 0.15}$ | $33.83_{\pm 0.45}$ | $21.46_{\pm 0.19}$ |
| MGAE | ✓ | ✗ | $47.83_{\pm 2.33}$ | $\mathbf{56.14}_{\pm 1.00}$ | $35.87_{\pm 0.97}$ | $30.57_{\pm 0.98}$ | $32.73_{\pm 1.16}$ | $27.95_{\pm 2.29}$ |
| ARGE | ✓ | ✗ | $42.04_{\pm 0.50}$ | $44.13_{\pm 0.91}$ | $36.07_{\pm 0.05}$ | $27.74_{\pm 0.01}$ | $26.49_{\pm 0.10}$ | $17.17_{\pm 0.05}$ |
| ARVGE | ✓ | ✗ | $21.10_{\pm 0.61}$ | $21.79_{\pm 0.49}$ | $26.45_{\pm 0.03}$ | $12.94_{\pm 0.01}$ | $33.82_{\pm 0.13}$ | $21.42_{\pm 0.11}$ |
| AGC | ✓ | ✗ | $38.83_{\pm 0.84}$ | $47.08_{\pm 1.57}$ | $\mathbf{41.76}_{\pm 0.01}$ | $\mathbf{33.65}_{\pm 0.01}$ | $32.74_{\pm 0.01}$ | $24.90_{\pm 0.01}$ |
| DSGC ($GX$) | ✓ | ✗ | $38.42_{\pm 0.66}$ | $46.28_{\pm 0.93}$ | $38.26_{\pm 0.02}$ | $30.66_{\pm 0.02}$ | $31.43_{\pm 0.09}$ | $24.16_{\pm 0.18}$ |
| DSGC ($XF$) | ✗ | Emb | $28.99_{\pm 0.06}$ | $33.22_{\pm 0.10}$ | $30.80_{\pm 0.56}$ | $17.46_{\pm 0.21}$ | $\mathbf{35.45}_{\pm 0.91}$ | $\mathbf{33.44}_{\pm 0.66}$ |
| | ✗ | PPMI | $48.36_{\pm 2.40}$ | $53.27_{\pm 2.17}$ | $36.46_{\pm 0.06}$ | $22.53_{\pm 0.03}$ | $\mathbf{38.10}_{\pm 0.01}$ | $\mathbf{36.07}_{\pm 0.02}$ |
| DSGC ($GXF$) | ✓ | Emb | $43.40_{\pm 0.66}$ | $50.97_{\pm 0.58}$ | $40.75_{\pm 0.02}$ | $\mathbf{33.05}_{\pm 0.04}$ | $30.50_{\pm 0.01}$ | $25.48_{\pm 0.03}$ |
| | ✓ | PPMI | $\mathbf{52.25}_{\pm 1.97}$ | $\mathbf{61.34}_{\pm 1.07}$ | $41.24_{\pm 0.04}$ | $30.92_{\pm 0.01}$ | $31.37_{\pm 0.08}$ | $26.06_{\pm 0.20}$ |

⋆ ✓ and ✗ indicate using/not using $G$ or $F$.

• As shown in Table 2, for node classification with 20 labels per class on all three datasets, after incorporating DSGC, the performances of GCN, GAT, LDS, and GraphSAGE are substantially improved in most cases. This further verifies the effectiveness of 2-D graph convolution.

## 7.2 Node Clustering

**Baselines.** We test the proposed node clustering method by DSGC (section 5.1) with or without using $G$ and $F$ in five cases. We compare DSGC with strong baselines for attributed graph clustering including GAE and VGAE (Kipf & Welling, 2016), MGAE (Wang et al., 2017), ARGE and ARVGE (Pan et al., 2018), and AGC (Zhang et al., 2019). We also compare with spectral clustering (Spectral) that operates on a similarity matrix constructed by applying linear kernel on the node representations.

**Performance.** To evaluate the clustering performance, we adopt two widely-used metrics (Aggarwal & Reddy, 2014): clustering accuracy (Acc) and normalized mutual information (NMI). The results are shown in Table 3, where the top 2 results are highlighted in bold. The following observations can be made. **1)** Attribute graph convolution works very well. On 20 NG, DSGC ($XF$) with PPMI already outperforms most baselines by a large margin. On Wiki, DSGC ($XF$) with PPMI or Emb significantly outperforms all the baselines. **2)** Similar to the classification tasks, 2-D graph convolution is effective. On 20 NG, DSGC ($GXF$) with PPMI can further improve upon the very strong performance of DSGC ($XF$) and performs the best; On L-Cora, DSGC ($GXF$) with PPMI or Emb improves upon either DSGC ($GX$) or DSGC ($XF$) and outperforms most baselines significantly. On Wiki, DSGC ($XF$) performs better than DSGC ($GXF$), due to the low-quality hyperlink graph as explained above.

For both classification and clustering, we observe that in most cases DSGC with PPMI can achieve better performance than with Emb. This shows the effectiveness of PPMI in capturing meaningful word relations based on information theory and statistics (Church & Hanks, 1989), whereas Emb only relies on a distance metric for measuring word similarity.

## 8 Conclusion

In this paper, we have proposed to model attributed graphs with 2-D graph convolution. We have demonstrated theoretically and empirically that by exploiting attribute relations in addition to node relations, a simple and efficient dimensionwise separable 2-D graph convolution (DSGC) can learn better node representations than existing methods based on regular 1-D graph convolution on noisy and sparse real-word networks. We believe 2-D graph convolution is a promising tool for attributed graph learning, and there is much left to be explored. In future work, we plan to further investigate the construction of attribute affinity graphs for different types of attributed networks, apply DSGC to solve various practical problems, and design new efficient filters for 2-D graph convolution.

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

# Appendices

Appendix A provides details about the experiments in section 7 and presents additional experimental results. Appendix B, C and D provide proofs of the theorems in section 4.

## APPENDIX A    EXPERIMENT DETAILS AND SUPPLEMENTARY EXPERIMENTS

### A.1    DATASET DETAILS

Table 4: Classes of each dataset.

| Datasets | Classes | | | |
|---|---|---|---|---|
| 20 NG | talk.politics.guns
talk.politics.mideast
talk.politics.misc
talk.religion.misc
soc.religion.christian | sci.crypt
sci.electronics
sci.med
sci.space
alt.atheism | rec.autos
rec.motorcycles
rec.sport.baseball
rec.sport.hockey
misc.forsale | comp.graphics
comp.os.ms-windows.misc
comp.sys.ibm.pc.hardware
comp.sys.mac.hardware
comp.windows.x |
| Wiki | Everyday Life
Countries
Citizenship | Religion
People | Science
History | Design and Technology
Language and literature |
| L-Cora | Artificial Intelligence
Data Structures Algorithms and Theory
Operating Systems
Information Retrieval | Networking

Programming
Databases | Encryption and Compression
Hardware and Architecture
Human Computer Interaction | |

**20 Newsgroups** (20 NG) Lang (1995) is an email discussion group, where each node is an email and there are 18846 emails in total. Each email is represented by an 11697-dimension tf-idf feature vector. Two emails are connected by an edge if they replies the same one. These emails are categorized into 20 classes as listed in Table 4.

**Wikispeedia** (Wiki) West et al. (2009); West & Leskovec (2012) is a webpage network in which the nodes are 3767 Wikipedia webpages, and the edges are web hyperlinks. Each webpage is described by a 18316-dimension tf-idf vector. We remove several tiny classes from the dataset, so the webpages distribute more evenly across the remaining 9 categories, which are also listed in Table 4.

**Large Cora** (L-Cora) McCallumzy et al. (1999) is a citation network in which the nodes are computer science research papers represented by 3780 dimension of tf-idf values. Two papers are connected by an undirected edge if and only if one cites the other. These citation links form a node graph. The nodes in the dataset are originally categorized into a topic hierarchical tree with 73 leaves. After removing the papers that belong to no topic and the ones that have no authors or title, a subset of 11881 papers is obtained Saccá et al. (2013). These papers are then classified into 10 highest-level topics in the topic hierarchy, as listed in Table 4. We name this dataset "Large Cora" to distinguish it from the "Cora" dataset with 2708 papers used in Kipf & Welling (2017); Yang et al. (2016); Sen et al. (2008). Note that we did not test on this "Cora" and the "Citeseer" and "PubMed" datasets as in Kipf & Welling (2017); Yang et al. (2016); Sen et al. (2008), because the attributes (words) are not provided in these datasets.

The statistics of the datasets are summarized in Table 5, where the last row shows the intra-class edge ratio of the node graph of each dataset, which can reflect the quality of the graph.

### A.2    VISUALIZATION

In Figure 3, we visualize the results of performing graph convolution on the node features of 20 NG dataset by t-SNE. It can be seen that graph convolution can successfully reduce the overlap among classes, and 2-D graph convolution is more effective than 1-D.

Table 5: Dataset statistics.

| Dataset | 20 NG | Wiki | L-Cora |
|---|---|---|---|
| Vertices | 18846 | 3767 | 11881 |
| Edges | 147034 | 129597 | 64898 |
| Classes | 20 | 9 | 10 |
| Features | 11697 | 18316 | 3780 |
| Connected Components | 8504 | 303 | 833 |
| Intra-class edge ratio | 96.8% | 38.0% | 76.5% |

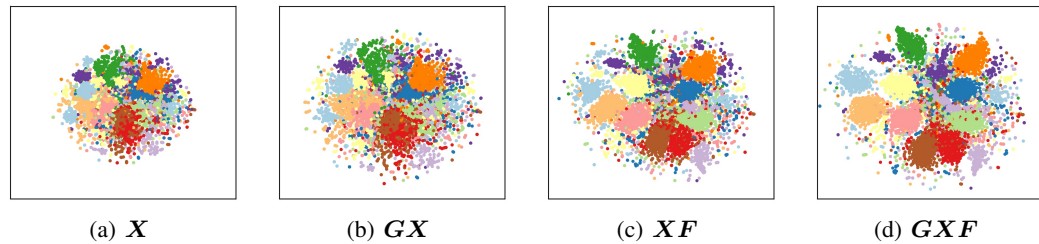

(a) $X$      (b) $GX$      (c) $XF$      (d) $GXF$

Figure 3: t-SNE visualization of "20 NG". (a) Raw features; (b) Results of node graph convolution; (c) Results of affinity graph convolution; (d) Results of 2-D graph convolution (DSGC).

### A.3 TRAINING TIME

The training time is summarized in Figure 4. We can see that DSGC is several times faster than GCN, GAT, DGI and GraphSAGE. This is because DSGC only performs graph convolution once before training, while others need to do graph convolution in every training step.

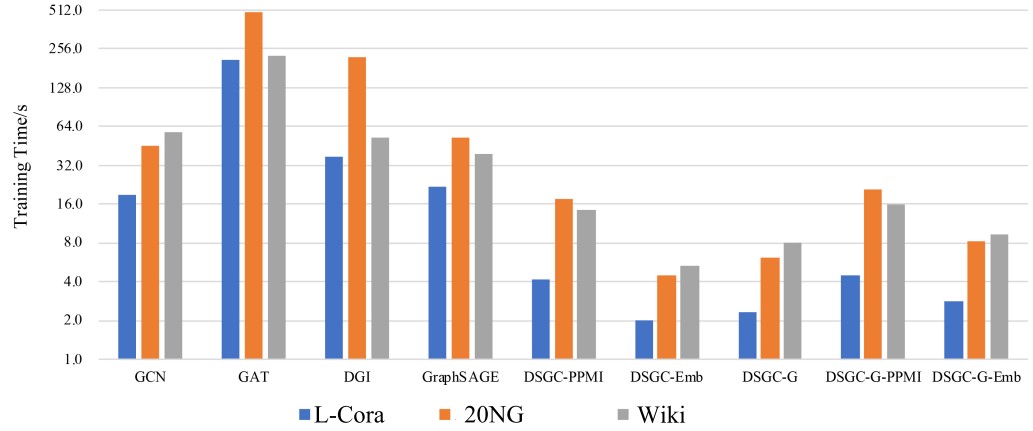

Figure 4: Training time of DSGC and baselines.

### A.4 PARAMETER SETTINGS FOR NODE CLASSIFICATION

For DSGC, we test positive point-wise mutual information (PPMI) and word embedding based $k$-NN graphs (Emb) for constructing the attribute affinity graph. For constructing the PPMI graph, we set the context window size to 20 and use the inverse distance as co-occurrence weight. For constructing the Emb graph, we use GloVe Pennington et al. (2014) to learn word embeddings and set the number of nearest neighbours $k = 20$. The classifier of DSGC is a two-layer multi-layer perceptron (MLP) with 64 hidden units. The MLP is trained with Adam Optimizer for 200 epochs with 0.015 learning rate. For the sake of fair comparison to all baselines and to make sure our method can work well in practice, we do not use extra labeled data for validation on our methods and the baselines, unless

otherwise stated. Instead, we always select the model with the lowest train loss in the 200 epochs. In each run, the dataset is randomly split into a labeled set and an unlabeled set. The classification results are averaged over 10 runs.

Hyperparameters of all models, including our methods and baselines, are tuned to its best performance by grid search. DSGC follows the network setting and hyperparameters of GCN, which are a 2-layer structure with 64 hidden units and 0.015 learning rate. We set the balancing parameter $\alpha$ in LP as 100. We largely follow Velickovic et al. (2018) to set network structures and hyperparameters of GAT, which are 2 layers, 8 heads, 8 neurons of each head, 0.005 learning rate, 400 epochs, and 0.3 dropout. Our setting of DGI is the same as Velickovic et al. (2019), except that the number of hidden units is decreased to 128, so it can be fitted into our GPU memory. Our setting of GraphSAGE is the same as Hamilton et al. (2017) (using 30% data as validation set), except that the number of samples in layer 2 is reduced to 5, also due to limited GPU memory. Parameters of GLP are same as Li et al. (2019). For LDS Franceschi et al. (2019), since it cannot scale to the size of 20 NG (out of GPU memory), we follow the authors to select a 10-category subset of 20 NG. For each dataset, both the training set and validation set for LDS contain 20 labels per class.

All models are tested on a platform with Intel(R) Xeon(R) CPU E5-2640 v4 @ 2.40GHz and single NVIDIA(R) GeForce(R) GTX 1080 Ti.

### A.5 PARAMETER SETTINGS FOR NODE CLUSTERING

For our method DSGC, the attribute affinity graphs PPMI and Emb are constructed in the same way as in node classification. For other baselines, we follow the parameter settings described in the original papers. In particular, for GAE and VGAE (Kipf & Welling, 2016), we construct encoders with a 32-neuron hidden layer and a 16-neuron embedding layer, and train the encoders for 200 iterations using the Adam algorithm with a learning rate of 0.01. For MGAE (Wang et al., 2017), the corruption level $p$ is 0.4, the number of layers is 3, and the parameter $\lambda$ is $10^{-5}$. For ARGE and ARVGE (Pan et al., 2018), we construct encoders with a 32-neuron hidden layer and a 16-neuron embedding layer. The discriminators are built by two hidden layers with 16 neurons and 64 neurons respectively. We train all the autoencoder-related models for 200 iterations and optimize them using the Adam algorithm. The learning rate and the discriminator learning rate are both 0.001. For AGC (Zhang et al., 2019), the maximum iteration number is 60. For fair comparison, these baselines also adopt spectral clustering as DSGC to get the clustering results. We repeat each method for 10 times, and show the mean clustering results and the standard deviations.

## APPENDIX B    PROOF OF THEOREM 1 IN SECTION 4.1

We suppose that nodes from the same class are connected with probability $p$, and nodes from different classes are connected with probability $q$, i.e., the adjacency matrix $\boldsymbol{A}^{(1)}$ of node graph obeys

$$\Pr(a_{ij} = 1) = \left\{ \begin{array}{ll} p, & \text{if } y_i = y_j \\ q, & \text{if } y_i \neq y_j \end{array} \right. , \quad \Pr(a_{ij} = 0) = \left\{ \begin{array}{ll} 1 - p, & \text{if } y_i = y_j \\ 1 - q, & \text{if } y_i \neq y_j \end{array} \right. \tag{15}$$

and $\boldsymbol{G} = \boldsymbol{D}^{-1}\boldsymbol{A}^{(1)}$ is a stochastic matrix. We also assumes that classes are balanced, i.e., $\Pr(\mathbb{Y} = k) = 1/K$ for all $k$, then we have following theorem.

**Theorem 1.** *When $q$ is sufficiently small, $\mathbb{Z}$ has an* IntraVar/InterVar *ratio less than or equal to that of $\mathbb{X}$, i.e.,*

$$\frac{\mathrm{E}\left[\mathrm{Var}\left(\mathbb{Z}|\mathbb{Y}\right)\right]}{\mathrm{Var}\left(\mathrm{E}\left[\mathbb{Z}|\mathbb{Y}\right]\right)} \leq \frac{\mathrm{E}\left[\mathrm{Var}\left(\mathbb{X}|\mathbb{Y}\right)\right]}{\mathrm{Var}\left(\mathrm{E}\left[\mathbb{X}|\mathbb{Y}\right]\right)}. \tag{16}$$

*Proof.* The proof consists of two parts. In the first part, we prove that inter-class variance is unchanged after node graph convolution, when $q$ approximates 0, i.e.,

$$\lim_{q \to 0} \mathrm{Var}\left(\mathrm{E}\left[\mathbb{Z}|\mathbb{Y}\right]\right) = \mathrm{Var}\left(\mathrm{E}\left[\mathbb{X}|\mathbb{Y}\right]\right). \tag{17}$$

In the second part, we prove that intra-class variance becomes smaller after node graph convolution, i.e.,

$$\mathrm{E}\left[\mathrm{Var}\left(\mathbb{Z}|\mathbb{Y}\right)\right] \leq \mathrm{E}\left[\mathrm{Var}\left(\mathbb{X}|\mathbb{Y}\right)\right], \tag{18}$$

when $G$ is a stochastic matrix.

**Part 1. Inter-class variance is unchanged.**    Since

$$\boldsymbol{z}_i = \sum_j G_{ij}\boldsymbol{x}_j,$$

we have

$$
\begin{aligned}
\mathrm{E}\left[\boldsymbol{z}_i|y_i = k\right] &= \sum_j \mathrm{E}\left[G_{ij}\right]\mathrm{E}\left[\boldsymbol{x}_j\right] && \text{\# by linearity of expectation} \\
&= \sum_{j, y_j = k} \mathrm{E}\left[G_{ij}\right]\mathrm{E}\left[\boldsymbol{x}_j\right] + \sum_{j, y_j \neq k} \mathrm{E}\left[G_{ij}\right]\mathrm{E}\left[\boldsymbol{x}_j\right] \\
&= \frac{\sum_{j, y_j = k}\mathrm{E}\left[a_{ij}\right]\mathrm{E}\left[\boldsymbol{x}_j\right] + \sum_{j, y_j \neq k}\mathrm{E}\left[a_{ij}\right]\mathrm{E}\left[\boldsymbol{x}_j\right]}{\sum_j \mathrm{E}\left[a_{ij}\right]} \\
&= \frac{p\sum_{j, y_j = k}\mathrm{E}\left[\mathbb{X}|\mathbb{Y} = k\right] + q\sum_{j, y_j \neq k}\mathrm{E}\left[\boldsymbol{x}_j\right]}{\frac{N}{K}(p - q) + Nq} \\
&= \frac{\frac{N}{K}p\mathrm{E}\left[\mathbb{X}|\mathbb{Y} = k\right] + q\sum_j \mathrm{E}\left[\boldsymbol{x}_j\right] - q\sum_{j, y_j = k}\mathrm{E}\left[\boldsymbol{x}_j\right]}{\frac{N}{K}(p - q) + Nq} \\
&= \frac{\frac{N}{K}(p - q)\mathrm{E}\left[\mathbb{X}|\mathbb{Y} = k\right] + Nq\mathrm{E}\left[\mathbb{X}\right]}{\frac{N}{K}(p - q) + Nq} \\
&= \frac{(p - q)\mathrm{E}\left[\mathbb{X}|\mathbb{Y} = k\right] + Kq\mathrm{E}\left[\mathbb{X}\right]}{(p - q) + Kq}
\end{aligned}
$$

When $q$ approximates 0, $\mathrm{E}\left[\boldsymbol{z}_i|y_i = k\right]$ will approximate $\mathrm{E}\left[\mathbb{X}|\mathbb{Y} = k\right]$, so

$$
\begin{aligned}
\mathrm{E}\left[\mathbb{Z}|\mathbb{Y} = k\right] &= \sum_{i, y_i = k} \Pr(\mathbb{Z} = \boldsymbol{z}_i|y_i = k)\mathrm{E}\left[\boldsymbol{z}_i|y_i = k\right] \\
&= \sum_{i, y_i = k} \Pr(\mathbb{Z} = \boldsymbol{z}_i|y_i = k)\mathrm{E}\left[\mathbb{X}|\mathbb{Y} = k\right] \\
&= \mathrm{E}\left[\mathbb{X}|\mathbb{Y} = k\right].
\end{aligned}
$$

Hence,

$$\mathrm{Var}\left(\mathrm{E}\left[\mathbb{Z}|\mathbb{Y}\right]\right) = \mathrm{Var}\left(\mathrm{E}\left[\mathbb{X}|\mathbb{Y}\right]\right). \tag{19}$$

**Part 2. Intra-class variance becomes smaller.** Denote by $\text{Cov}(\cdot, \cdot)$ the covariance of two random variables. We have

$$
\begin{aligned}
\text{Var}\left(\sum_j G_{ij}\boldsymbol{x}_j\right) &= \sum_j G_{ij}^2 \text{Var}\left(\boldsymbol{x}_j\right) + \sum_{j,l} G_{ij}G_{il}\text{Cov}\left(\boldsymbol{x}_j, \boldsymbol{x}_l\right) && \text{\# property of variance} \\
&\leq \sum_{j,l} G_{ij}G_{il}\sqrt{\text{Var}\left(\boldsymbol{x}_j\right)}\sqrt{\text{Var}\left(\boldsymbol{x}_l\right)} && \text{\# property of covariance} \\
&= \left(\sum_j G_{ij}\sqrt{\text{Var}\left(\boldsymbol{x}_j\right)}\right)^2.
\end{aligned}
$$

Since

$$
\begin{aligned}
\text{Var}\left(\boldsymbol{z}_i | y_i = k\right) &= \text{Var}\left(\sum_j G_{ij}\boldsymbol{x}_j \,\middle|\, y_j = k\right) \\
&\leq \left(\sum_j G_{ij}\sqrt{\text{Var}\left(\boldsymbol{x}_j | y_j = k\right)}\right)^2 && \text{\# by the above inequality} \\
&= \left(\sum_j G_{ij}\sqrt{\text{Var}\left(\mathbb{X} | \mathbb{Y} = k\right)}\right)^2 \\
&= \left(\sqrt{\text{Var}\left(\mathbb{X} | \mathbb{Y} = k\right)}\right)^2 && \text{\# since } \sum_j G_{ij} = 1 \\
&= \text{Var}\left(\mathbb{X} | \mathbb{Y} = k\right),
\end{aligned}
$$

we have

$$
\text{Var}\left(\mathbb{Z} | \mathbb{Y} = k\right) = \sum_{i, y_i = k} \text{Pr}(\mathbb{Z} = \boldsymbol{z}_i | y_i = k)\text{Var}\left(\boldsymbol{z}_i | y_i = k\right) \leq \text{Var}\left(\mathbb{X} | \mathbb{Y} = k\right). \tag{20}
$$

Then we have

$$
\begin{aligned}
\text{E}\left[\text{Var}\left(\mathbb{Z} | \mathbb{Y}\right)\right] &= \sum_k \text{Pr}(\mathbb{Y} = k)\text{Var}\left(\mathbb{Z} | \mathbb{Y} = k\right) \\
&\leq \sum_k \text{Pr}(\mathbb{Y} = k)\text{Var}\left(\mathbb{X} | \mathbb{Y} = k\right) && \text{\# by the above inequality} \\
&= \text{E}\left[\text{Var}\left(\mathbb{X} | \mathbb{Y}\right)\right]. \tag{21}
\end{aligned}
$$

Combining Eq. (19) and Eq. (21), we prove that when $q$ is sufficiently small,

$$
\frac{\text{E}\left[\text{Var}\left(\mathbb{Z} | \mathbb{Y}\right)\right]}{\text{Var}\left(\text{E}\left[\mathbb{Z} | \mathbb{Y}\right]\right)} \leq \frac{\text{E}\left[\text{Var}\left(\mathbb{X} | \mathbb{Y}\right)\right]}{\text{Var}\left(\text{E}\left[\mathbb{X} | \mathbb{Y}\right]\right)}. \tag{22}
$$

$\square$

## APPENDIX C    PROOF OF THEOREM 2 IN SECTION 4.2

**Theorem 2.** *If the attribute graph convolutional filter $\boldsymbol{F}$ is a doubly stochastic matrix, then the output of attribute graph convolution has an intra-class variance less than or equal to that of $\mathbb{X}$, i.e.,*

$$\sum_i F_{ij} = \sum_j F_{ij} = 1 \text{ and } F_{ij} \geq 0, \forall\, i, j \quad \Rightarrow \quad \mathrm{E}\left[\mathrm{Var}\left(\boldsymbol{F}^\top\mathbb{X}|\mathbb{Y}\right)\right] \leq \mathrm{E}\left[\mathrm{Var}\left(\mathbb{X}|\mathbb{Y}\right)\right].$$

*Proof.* We first prove a lemma that variance of each class will not increase after attribute graph convolution, i.e., $\mathrm{Var}\left(\boldsymbol{F}^\top\mathbb{X}|\mathbb{Y} = k\right) \leq \mathrm{Var}\left(\mathbb{X}|\mathbb{Y} = k\right)$. Denote by $\mathrm{Cov}\left(\cdot\right)$ the covariance matrix of a random vector. Based on our definition of variance in section 4, we have

$\mathrm{Var}\left(\boldsymbol{F}^\top\mathbb{X}|\mathbb{Y} = k\right) = \mathrm{Tr}\left(\mathrm{Cov}\left(\boldsymbol{F}^\top\mathbb{X}|\mathbb{Y} = k\right)\right)$

$$\begin{aligned}
&= \mathrm{Tr}\left(\boldsymbol{F}^\top\mathrm{Cov}\left(\mathbb{X}|\mathbb{Y} = k\right)\boldsymbol{F}\right) && \text{\# property of covariance} \\
&= \mathrm{Tr}\left(\mathrm{Cov}\left(\mathbb{X}|\mathbb{Y} = k\right)\boldsymbol{F}\boldsymbol{F}^\top\right) && \text{\# cyclic property of trace} \\
&= \sum_{ij}\mathrm{Cov}\left(\mathbb{X}_i, \mathbb{X}_j|\mathbb{Y} = k\right)(\boldsymbol{F}\boldsymbol{F}^\top)_{ij} && \text{\# property of trace} \\
&= \sum_{ij}\mathrm{Cov}_{ij}^{(k)}(\boldsymbol{F}\boldsymbol{F}^\top)_{ij} && \\
&\leq \sum_{ij}\sqrt{\mathrm{Var}\left(\mathbb{X}_i|\mathbb{Y} = k\right)}\sqrt{\mathrm{Var}\left(\mathbb{X}_j|\mathbb{Y} = k\right)}(\boldsymbol{F}\boldsymbol{F}^\top)_{ij} && \text{\# property of covariance} \\
&= \sum_{ij}\sigma_i\sigma_j(\boldsymbol{F}\boldsymbol{F}^\top)_{ij} && \text{\# } \boldsymbol{\sigma} \in \mathbb{R}^m, \sigma_i \triangleq \sqrt{\mathrm{Var}\left(\mathbb{X}_i|\mathbb{Y} = k\right)} \\
&= \boldsymbol{\sigma}^\top\boldsymbol{F}\boldsymbol{F}^\top\boldsymbol{\sigma} && \\
&\leq \|\boldsymbol{\sigma}\|_2^2 && \text{\# eigenvalues of } \boldsymbol{F} \text{ is no more than 1} \\
&= \sum_i\mathrm{Var}\left(\mathbb{X}_i|\mathbb{Y} = k\right) && \\
&= \mathrm{Var}\left(\mathbb{X}|\mathbb{Y} = k\right). &&
\end{aligned}$$

Next, we prove the theorem with the above lemma. Denote by $\pi_k = \mathrm{Pr}(\mathbb{Y} = k)$ the portion of each class, then we have

$$\begin{aligned}
\mathrm{E}\left[\mathrm{Var}\left(\boldsymbol{F}^\top\mathbb{X}|\mathbb{Y}\right)\right] &= \sum_k\pi_k\mathrm{Var}\left(\boldsymbol{F}^\top\mathbb{X}|\mathbb{Y} = k\right) \\
&= \sum_k\pi_k\sum_{ij}\mathrm{Cov}_{ij}^{(k)}(\boldsymbol{F}\boldsymbol{F}^\top)_{ij} \\
&\leq \sum_k\pi_k\mathrm{Var}\left(\mathbb{X}|\mathbb{Y} = k\right) \\
&= \mathrm{E}\left[\mathrm{Var}\left(\mathbb{X}|\mathbb{Y}\right)\right]
\end{aligned}$$

$\square$

**Construction of a Doubly Stochastic Filter $\boldsymbol{F}$**    Given an attribute affinity matrix $\boldsymbol{A}^{(2)}$, one could easily construct a doubly stochastic filter $\boldsymbol{F}$ by the following steps: 1) compute graph Laplacian $\boldsymbol{L}^{(2)} = \boldsymbol{D}^{(2)} - \boldsymbol{A}^{(2)}$; 2) compute doubly stochastic graph matrix $\boldsymbol{\Omega} = (\boldsymbol{I} + \boldsymbol{L}^{(2)})^{-1}$; 3) take $\boldsymbol{\Omega}$ as a new attribute affinity matrix, choose a polynomial $p$ with nonnegative coefficients that sum to 1, and let $\boldsymbol{F} = p(\boldsymbol{\Omega})$. It can be easily seen that $\boldsymbol{F}$ is doubly stochastic.

## APPENDIX D   PROOF OF THEOREM 3 IN SECTION 4.2

**Theorem 3.** *If $\forall F_{ij} \neq 0$, $\|e_i - e_j\|_2 \leq \varepsilon$, then the distance between $e_j$ and $\widehat{e}_j = \sum_i F_{ij} e_i$ is also less than or equal to $\varepsilon$, i.e.,*

$$\|e_i - e_j\|_2 \leq \varepsilon, \ \forall F_{ij} \neq 0 \quad \Rightarrow \quad \|e_j - \widehat{e}_j\|_2 \leq \varepsilon,$$

*and $\varepsilon$ can be arbitrarily small with a proper $\boldsymbol{F}$.*

*Proof.*

$$\begin{aligned}
\|e_j - \widehat{e}_j\|_2 &= \left\| e_j - \sum_i F_{ij} e_i \right\|_2 \\
&= \left\| \sum_i F_{ij} (e_j - e_i) \right\|_2 & \# \text{ since } \sum_i F_{ij} = 1 \\
&\leq \sum_i F_{ij} \|e_j - e_i\|_2 & \# \text{ Cauchy-Schwarz inequality} \\
&\leq \sum_i F_{ij} \varepsilon = \varepsilon
\end{aligned}$$

Next, we prove that there exists such an $F$ that $\varepsilon$ is 0. This is equivalent to finding a doubly stochastic $F$ satisfying $\sum_i F_{ij} e_i = e_j$ for all $j$. Since $F = I$ is a trivial solution, it is solvable. Denote by $m$ the number of attributes, and denote by $K$ the number of classes. This linear system consists of $m(K + 2)$ equations and $m^2$ variables. In most real-world attributed networks, the number of attributes is far greater than the number of classes, so the number of variables in this linear system is greater than the number of equations. Given that it is solvable, it must have infinite number of solutions other than $I$. Thus, $\varepsilon$ can be arbitrarily small with a proper $\boldsymbol{F}$. $\qquad \square$

