# OpenReview forum: "Attributed Graph Learning with 2-D Graph Convolution"
_ICLR.cc/2020/Conference — Reject_

### Official Review · AnonReviewer1 · 2019-10-17
**Official Blind Review #1**

**Rating:** 3

**Review:**

The authors propose a method that incorporates two types of graphs into a graph convolutional networks:

(1) the given graph, which the authors refer to as node affinity graph, and
(2) the graph that models an affinity between node attribute values.

The main contribution of their work is a type of graph convolution that combines convolutions operating on these two graphs.

The paper is densely written and provides several mathematical derivations that are unnecessary to convey the proposed method. Personally, I don't see any benefits in having sections 3.1-3.4 in the main paper. The method actually proposed and evaluated in the paper is described in section 3.5. Sections 3.1-3.4 could be moved to an appendix. They confuse the reader more than they help. (They demonstrate knowledge of graph signal processing on the parts of the authors but little more.)

Trying to provide some theoretical analysis of the proposed method (and standard graph convolutions) by showing that the intra-class variance is reduced is laudable. The theorems, however, only hold under strong assumptions and could, in my opinion, also be moved to an appendix. In the end, they don't have any bearing on the performance of the methods using real-world datasets. Adding some experiments to analyse to what extent the assumptions made by the theorems are met in the given datasets would be an interesting addition to the paper.

The authors discuss related work sufficiently with one exception: there has been recent work on learning the structure of graph neural networks. See for example [1]. The structure is derived/bootstrapped using node attribute similarities and it is shown that augmenting the graph with these new edges improves accuracy significantly. I would like to point the authors specifically to Figure 2 and Table 1 in said paper, where the authors show that adding edges (e.g., based on some node attribute affinity before or during training) is beneficial and improves accuracy. It would therefore be interesting to see how the authors proposed 2-D convolution would compare to a baseline where the edges based on attribute affinity are added to the original (node affinity) graph. It is a (somewhat simpler) alternative way to combine node affinity and node attribute graphs.

[1] https://arxiv.org/pdf/1903.11960.pdf

The empirical results are mixed. Due to the numerous different variations of DSGC for which experiments were conducted, the difference between DSGC and existing methods is probably not statistically significant (a bonferroni correction was not performed to counteract the multiple comparisons).

Overall this an interesting paper that introduces a way to incorporate node attribute affinity graphs. It is too densely written and could benefit from moving the theoretical parts to an appendix. They don't really add much to the core of the paper. Moreover, the authors do not consider approaches that also add edges to the graph (based, e.g., on attribute value similarity or during learning, see e.g. [1]) showing that that improves performance even when using a vanilla GCN. A comparison to a baseline that simply adds edges based on attribute affinity to the graph and applied a vanilla GCN should be part of the evaluation. The empirical results are mixed and don't show a clear advantage of the proposed method.


**Experience Assessment:**

I have published in this field for several years.

**Review Assessment: Checking Correctness Of Derivations And Theory:**

I assessed the sensibility of the derivations and theory.

**Review Assessment: Checking Correctness Of Experiments:**

I carefully checked the experiments.

**Review Assessment: Thoroughness In Paper Reading:**

I read the paper at least twice and used my best judgement in assessing the paper.

---

> ### Author Response · Authors · 2019-11-15
> **Response to AnonReviewer1**
>
> Thank you for your positive and constructive feedback.
>
> Q1. Comparison with works on learning graph structures such as LDS (https://arxiv.org/pdf/1903.11960.pdf).
>
> >> First of all, we would like to thank the reviewer for pointing out this interesting work. In the revised manuscript, we have included some discussion on this line of research in the 3rd paragraph of section 2.
>
> We have also conducted experiments on LDS. Since LDS cannot scale to the size of the 20 Newsgroup dataset (out of GPU Memory) used in our experiments, we follow the authors to test on a 10-category subset of the 20 NG. We then test LDS on this subset of 20 NG, L-Cora, and Wiki. For classification on each dataset, LDS uses 20 labels per class for training and extra 20 labels per class for validation (the algorithm requires validation). Note that we do not use any validation data for the proposed DSGC method for classification. Due to the differences in datasets and experimental setup, we do not include the results of LDS in Table 1.
>
> Instead, we report the results of LDS in Table 2 to see whether the proposed DSGC can be used to improve LDS. We incorporate DSGC into LDS as described in section 5.2 by applying attribute graph convolution on the node features before training. The results in Table 2 show that DSGC significantly improves LDS on Newsgroup and Wiki and slightly improves LDS on L-Cora. We have also tested another case of LDS without using the given node affinity graphs of the three datasets and observed similar results.
>
> The experiments show that DSGC can complement and improve LDS, just as it can complement and improve other SOTA methods based on the regular 1-D graph convolution such as GCN/GAT/GraphSAGE as shown in Table 2.
>
>
> Q2. The paper is densely written.
>
> >> As the reviewer suggested, we have reorganized sections 3 and 4 to make them more compact in the revised manuscript. In section 3, we intend to show how the proposed 2-D graph convolution DSGC is derived, which follows a similar path of the development of 1-D GCN (from “spectral networks” to “ChebyNet” to “GCN”). In section 4, we want to provide some insights into why DSGC works by analyzing the variance reduction effect of node graph convolution and attribute graph convolution respectively.
>
> Q3. The empirical results are mixed.
>
> >> We have improved the presentation of the experiments in the revised manuscript. We kindly ask the reviewer to read section 7 about the experiments again. Our results are statistically significant. For datasets with good node affinity graphs such as 20 Newsgroup and L-Cora, the proposed 2-D graph convolution DSGC (GXF) significantly outperforms most SOTA methods. For datasets with bad node affinity graphs such as Wiki, the proposed 2-D graph convolution DSGC (GXF) still outperforms most SOTA methods by a large margin but is less effective than DSGC (XF) (since the node affinity graph G is bad). DSGC can also be used to significantly improve SOTA methods including GCN, GAT, LDS and GraphSAGE. Please refer to section 7 in the manuscript for more detailed explanation.

---

### Official Review · AnonReviewer2 · 2019-10-27
**Official Blind Review #3**

**Rating:** 6

**Review:**

This work proposes a 2D graph convolution to combine the relational information encoded both in the nodes and in the edges.

The basic idea is to reduce the intra-class variance. The authors provide theorems and proofs to support this claim even though it is quite intuitive that smoothing with similar neighbours preserves higher variance with respect to dissimilar neighbours.

It is not straightforward to understand the limitations on the size of graphs.

The experimental analysis provides the empirical evidence of the properties of the proposed method. It is worthwhile to remark that connectivity is not necessarily helpful, like in the wiki dataset where connected nodes are not similar.


**Experience Assessment:**

I have read many papers in this area.

**Review Assessment: Checking Correctness Of Derivations And Theory:**

I assessed the sensibility of the derivations and theory.

**Review Assessment: Checking Correctness Of Experiments:**

I assessed the sensibility of the experiments.

**Review Assessment: Thoroughness In Paper Reading:**

I read the paper at least twice and used my best judgement in assessing the paper.

---

> ### Author Response · Authors · 2019-11-15
> **Response to AnonReviewer2**
>
> Thank you for your positive and helpful feedback. As you suggested, we have further emphasized in section 7.2 of the revised manuscript that connectivity such as the hyperlinks in Wiki is not necessarily helpful.

---

### Official Review · AnonReviewer3 · 2019-10-28
**Official Blind Review #3**

**Rating:** 6

**Review:**

The paper proposes a new 2-D graph convolution method to aggregate information using both the node relation graph and the attribute graph generated using, e.g., PMI and KNN. The motivation does make sense that using 1-D convolution along the node dimension might not enough for learning representation for those low-degree nodes. Then, the attribute relation might be used to further smooth the node representation. The assumption could be that documents in a class are likely to consist of similar (related) words. To achieve this, the information aggregation along the node and the attribute dimension is implemented via a product of three matrices, the node graph convolutional filter computed from node affinity matrix, the attribute graph convolutional filter computed the attribute affinity matrix given by PMI or KNN, and the node-attribute matrix, which can be one main contribution.
Besides, the paper also includes a detailed discussion of intra-class variance reduction. They evaluated the proposed method on both the mode classification and the node clustering against several existing methods, demonstrated that the proposed method almost always outperforms those methods on the two datasets. Overall, it is an interesting paper.

For aggregating the information along the node dimension and the attribute dimension, as mentioned in the paper, is it possible to firstly do the propagation over the attribute graph using a 1-layer GCN, followed by another 1-layer GCN over the node relation graph, similar to dense graph propagation module in “Rethinking knowledge graph propagation for zero-shot learning”?

There are two ways to build an attribute graph. The experiments seem to show that the performance is quite different. It would be good to have some discussion on this.

One motivation is about the low-degree nodes, where the attribute graph might help. It would be good to have a study on the performance of the methods on those low-degree nodes.

**Experience Assessment:**

I have read many papers in this area.

**Review Assessment: Checking Correctness Of Derivations And Theory:**

I did not assess the derivations or theory.

**Review Assessment: Checking Correctness Of Experiments:**

I assessed the sensibility of the experiments.

**Review Assessment: Thoroughness In Paper Reading:**

I made a quick assessment of this paper.

---

> ### Author Response · Authors · 2019-11-15
> **Response to AnonReviewer3**
>
> Thank you for your positive and helpful comments.
>
> Q1. “Is it possible to firstly do the propagation over the attribute graph using a 1-layer GCN, followed by another 1-layer GCN over the node relation graph, similar to dense graph propagation module in “Rethinking knowledge graph propagation for zero-shot learning”?
>
> >> Yes, it is possible to do that, and the performance is expected to be similar as the proposed DSGC.
>
> Q2. Discussion on the two ways for constructing the attribute affinity graph.
>
> >>Thank you for the suggestion.  For both classiﬁcation and clustering, we observe that in most cases DSGC with PPMI can achieve better performance than with Emb. This shows the effectiveness of PPMI in capturing meaningful word relations based on information theory and statistics (Church & Hanks, 1989), whereas Emb only relies on a distance metric for measuring word similarity. We have also revised the manuscript to include discussion on this.
>
>
> Q3. “One motivation is about the low-degree nodes, where the attribute graph might help. It would be good to have a study on the performance of the methods on those low-degree nodes.
>
> >>This is a good point. Actually, we already did that. In our experiments, we compared the proposed attributed graph convolution DSGC (XF) with MLP. The former outperforms the latter by a very large margin on all the three datasets. Note that this is an extreme case where each node has 0 degree (GCN reduces to MLP in this case), which shows that attribute graph convolution works well even when there are no links between nodes. We have revised the manuscript to emphasize this point in section 7.2 as you suggested.

---

### Author Response · Authors · 2019-11-15
**To All: Manuscript Update**

We would like to thank all the reviewers for their valuable time and feedback.

We have incorporated their suggestion and revised the manuscript accordingly. Major changes include: 1) In section 7, we improve the presentation of experimental results and include comparison with a suggested baseline LDS; 2) We reorganize the content of section 3 and make section 4 more compact; 3) In section 2, we add some discussion of recent related work on learning graph structures for graph neural networks.

We will release source code and datasets to ensure the reproducibility of our results.

---

### Decision · Program_Chairs · 2019-12-19

**Decision:**

Reject

**Comment:**

The paper studies the problem of graph learning with attributes, and propose a 2-D graph convolution that models the node relation graph and the attribute graph jointly. The paper proposes and efficient algorithm and models intra-class variation. Empirical performance on 20-NG, L-Cora, and Wiki show the promise of the approach.

The authors responded to the reviews by updating the paper, but the reviewers unfortunately did not further engage during the discussion period. Therefore it is unclear whether their concerns have been adequately addressed.

Overall, there have been many strong submissions on graph neural networks at ICLR this year, and this submission as is currently stands does not quite make the threshold of acceptance.